# Towards Minimal Causal Representations for Human Multimodal Language Understanding

## Abstract

Human Multimodal Language Understanding (MLU) aims to infer human intentions by integrating related cues from heterogeneous modalities. Existing works predominantly follow a "learning to attend" paradigm, which maximizes mutual information between data and labels to enhance predictive performance. However, such methods are vulnerable to unintended dataset biases, causing models to conflate statistical shortcuts with genuine causal features and resulting in degraded out-of-distribution (OOD) generalization. To alleviate this issue, we introduce a **Ca**usal **M**ultimodal **I**nformation **B**ottleneck (CaMIB) model that leverages causal principles rather than traditional likelihood. Concretely, we first applies the information bottleneck to filter unimodal inputs, removing task-irrelevant noise. A parameterized mask generator then disentangles the fused multimodal representation into causal and shortcut subrepresentations. To ensure global consistency of causal features, we incorporate an instrumental variable constraint, and further adopt backdoor adjustment by randomly recombining causal and shortcut features to stabilize causal estimation. Extensive experiments on multimodal sentiment analysis, humor detection, and sarcasm detection, along with OOD test sets, demonstrate the effectiveness of CaMIB. Theoretical and empirical analyses further highlight its interpretability and soundness.

## 1 Introduction

Human Multimodal Language Understanding (MLU) aims to integrate diverse modalities—such as visual gestures, acoustic behaviors, linguistic texts, and physiological signals—to enable high-level semantic analysis of users' emotional states, making it a key technology for human–computer interaction (Xu et al., 2025). With the development of multimodal benchmarks (Zadeh et al., 2016; 2018; Hasan et al., 2019; Castro et al., 2019), numerous methods have been proposed to enhance model performance (Tsai et al., 2019; Rahman et al., 2020; Hasan et al., 2021; Yu et al., 2021; Li et al., 2023; Feng et al., 2024; Wang et al., 2025; Wu et al., 2025b). These approaches often rely on complex architectures or sophisticated fusion strategies. While effective on training datasets, they tend to produce high-dimensional embeddings that contain redundant information, which leads models to capture spurious correlations between inputs and labels, including dataset-specific biases and noise (Sun et al., 2022; Yang et al., 2024; Huang et al., 2025). As a result, their out-of-distribution (OOD) generalization deteriorates sharply. Subsection 5.3 presents experimental results that further illustrate the prevalence and severity of this issue.

Ideally, multimodal embeddings should satisfy two criteria: **i**) they capture the causal information necessary for prediction rather than relying on superficial statistical shortcuts, and **ii**) they minimize redundant information irrelevant to the prediction. However, achieving such ideal representations remains challenging. Information theory provides a principled framework for this purpose, with the Information Bottleneck (IB) method formalizing the objective through Mutual Information (MI): it maximizes the MI between the encoded representation and the labels while minimizing the MI between the representation and the inputs (Tishby et al., 2000). The core idea of IB is to quantify the complexity of input signals from an information-theoretic perspective, aiming to produce compact representations that retain predictive power while suppressing noise and redundancy (Mai et al., 2023c; Xiao et al., 2024; Wu et al., 2025a). In multimodal settings, however, merely maximizing MI can inadvertently amplify spurious correlations (Jiang et al., 2025). Unlike biased tendencies in

unimodal tasks, multimodal tasks often involve shared labels across modalities, causing models to entangle spurious signals from different modalities during representation learning. This entanglement can contaminate the learned representations with potential side effects (Yang et al., 2024).

Fortunately, causal inference (Pearl, 2009) provides a promising avenue for addressing this challenge, as it enables the identification of underlying causal relationships even in biased observational data. However, the effective application of causal inference to MLU tasks faces two major challenges. **i**) *How can causal and shortcut substructures be reliably identified in biased datasets?* When the test distribution deviates substantially from the training distribution, models tend to capture and exploit spurious correlations, which can lead to misleading predictions (Sui et al., 2022). Existing causal methods typically tackle this issue by explicitly defining specific bias types and mitigating them through counterfactual reasoning (Sun et al., 2022; Yang et al., 2024; Huan et al., 2024) or causal interventions (Xu et al., 2025; Jiang et al., 2025). Nevertheless, these approaches generally focus on local or narrowly defined bias patterns and lack the capacity to distinguish causal substructures from shortcut ones on a global scale. **ii**) *How can causal substructures be extracted from entangled multimodal inputs?* Statistically, causal substructures are determined by the global properties of multimodal inputs rather than by individual modalities or local features alone (Fan et al., 2022). Accurately capturing them therefore requires modeling both complex inter-modal interactions and intra-modal contextual dependencies.

In this paper, we first design a Structural Causal Model (SCM) tailored for MLU, which formalizes spurious correlations arising from redundant information as confounders, rather than restricting them to specific bias types. These confounders may mislead the model during inference, leading to biased predictions. Building on this foundation, we propose the **Ca**usal **M**ultimodal **I**nformation **B**ottleneck (CaMIB) model to mitigate confounding effects. Given multimodal inputs, we first apply the IB to filter out unimodal noise that is irrelevant to prediction. Next, we design a parameterized mask generator that partitions the fused multimodal representation into causal and shortcut components, with shared parameters across the representation space. To further reinforce disentanglement, we introduce an instrumental variable mechanism that leverages self-attention to capture inter-modal and token-level dependencies while ensuring global causal consistency. Finally, we adopt a backdoor adjustment strategy that randomly recombines causal and shortcut features to generate stratified samples with weakened correlations. Training on these samples encourages the model to prioritize causal over shortcut representations, thereby improving robustness under distributional shifts. Our main contributions are summarized as follows:

- We design a SCM tailored for MLU, which formalizes spurious correlations in redundant information as confounders. These confounders can mislead the model during inference, resulting in biased predictions.

- We propose a novel debiasing model, CaMIB, which integrates the IB principle with causal theory to fully exploit global causal features while effectively filtering out trivial patterns in multimodal inputs.

- Extensive experiments on multiple MLU tasks (including multimodal sentiment analysis, humor detection, and sarcasm detection) as well as on OOD test sets demonstrate that CaMIB outperforms existing methods, with particularly notable improvements under distribution shifts. Further analyses confirm the interpretability and soundness of CaMIB.

## 2 RELATED WORK

### 2.1 INFORMATION BOTTLENECK

IB provides a principled framework for learning compact representations that preserve task-relevant information (Tishby et al., 2000), and it was first applied to deep learning by Tishby & Zaslavsky (2015). Subsequently, the Variational Information Bottleneck (VIB) (Alemi et al., 2017) bridged IB and deep learning, enabling efficient approximation through stochastic variational inference. Recently, IB has been explored across diverse domains, including computer vision (Tian et al., 2021), reinforcement learning (Goyal et al., 2019), and natural language processing (Wang et al., 2020). In multimodal learning, approaches such as Multimodal Information Bottleneck (MIB) (Mai et al., 2023c) aim to learn effective multimodal representations by maximizing task-relevant information while reducing redundancy and noise. They also investigate the impact of applying IB at different

stages of modality fusion, which results in variants such as E-MIB, L-MIB, and C-MIB. In contrast, ITHP (Xiao et al., 2024) treats a dominant modality as the primary source and uses other modalities as auxiliary probes to capture complementary information. Although these methods achieve strong benchmark performance, they typically maximize MI between multimodal inputs and labels without discrimination. As a result, they may overlook dataset biases and inadvertently capture spurious correlations. By contrast, CaMIB provides a causal approach to address this limitation.

## 2.2 CAUSAL INFERENCE IN MULTIMODAL LEARNING

In recent years, causal inference has gained increasing attention in deep learning, aiming to identify and eliminate spurious correlations in complex data to enhance model robustness and generalization. Significant progress has been made in domains such as visual question answering (Niu et al., 2021), visual commonsense reasoning (Zhang et al., 2021), recommendation systems (Wang et al., 2022), and text classification (Qian et al., 2021). In multimodal learning, causal techniques have been explored to mitigate biases across modalities. Examples include counterfactual attention mechanisms for constructing more reliable attention distributions (Huang et al., 2025), front-door and back-door adjustments to remove spurious correlations between textual and visual modalities (Liu et al., 2023), and counterfactual frameworks (Sun et al., 2022; 2023; Yang et al., 2024; Huan et al., 2024), as well as generalized mean absolute error loss (Sun et al., 2023), both aiming to reduce spurious correlations within single modalities. Additionally, causal intervention modules have been designed to disentangle misleading associations between expressive styles and feature semantics (Xu et al., 2025), as well as to address both intra- and inter-modal biases (Jiang et al., 2025). Despite these advances, existing methods have notable limitations. Most approaches either focus on single modalities or specific modality pairs, or require explicitly labeled bias types, which demand extensive domain expertise (Nam et al., 2020). Consequently, their applicability to complex multimodal data with implicitly defined biases is constrained. In contrast, we propose a general and flexible debiasing approach that performs causal interventions directly on fused multimodal representations without requiring predefined bias types, thereby enhancing the generalization and applicability of models in complex multimodal scenarios.

## 3 CAUSAL ANALYSIS

To investigate the causal relationship between multimodal representation generation in MLU and task-specific predictions, we formalize interactions among four variables as a SCM: unobserved causal variables $C$, unobserved shortcut variables $Z$, multimodal representations $M$, and labels/predictions $Y$ (Figure 1). Each link encodes a causal dependency: **(1) Link** $C \rightarrow M \leftarrow Z$: $M$ is generated from $C$ and $Z$ (e.g., $C$ represents the animal, while $Z$ captures the background); **(2) Link** $C \leftrightarrow Z$: $C$ and $Z$ are entangled due to unobserved confounders; **(3)**

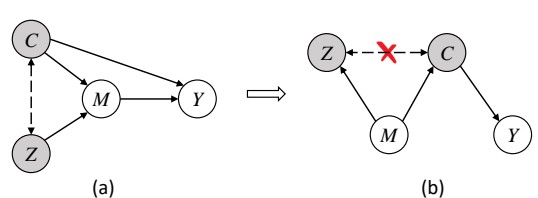

(a)          (b)

Figure 1: (a) SCM of multimodal representations in existing methods. (b) SCM of CaMIB.

**Link** $C \rightarrow Y$: $C$ is the only endogenous parent of $Y$; **(4) Link** $M \rightarrow Y$: existing MLU methods predict directly from $M$, potentially introducing spurious correlations caused by $Z$.

According to $d$-connection theory (Pearl, 2009), two variables are dependent if at least one unblocked path exists. As Figure 1(a) shows, $Z$ and $Y$ are connected via two unblocked paths: **i) Link** $Z \rightarrow M \rightarrow Y$ and **ii) Link** $Z \leftrightarrow C \rightarrow Y$, both inducing spurious correlations. Debiasing thus requires blocking both paths. Our approach (Figure 1(b)) is twofold: for path **i)**, we disentangle $C$ and $Z$ in $M$ and use only $C$ for prediction; for path **ii)**, since $C \rightarrow Y$ is immutable, we enforce independence between $C$ and $Z$ during learning, thereby effectively blocking the $C \leftrightarrow Z$ connection (red cross in the figure).

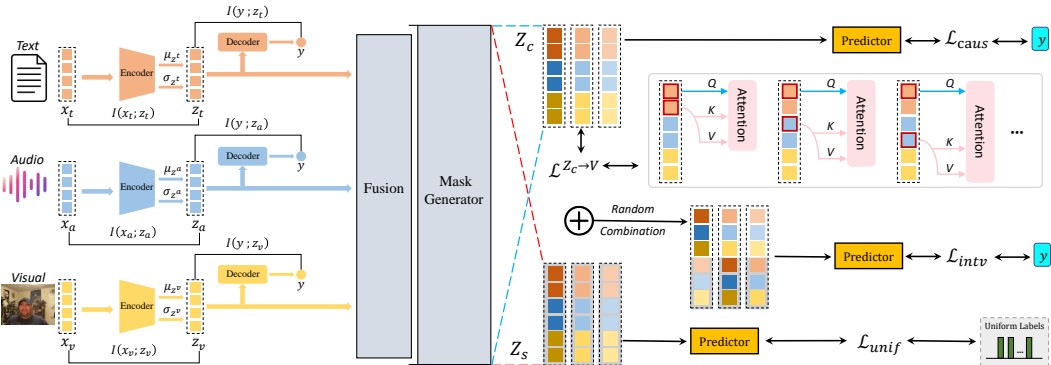

Figure 2: The overall framework of CaMIB, illustrated here using three modalities as an example.

## 4 METHODOLOGY

Motivated by the above causal analysis, we propose CaMIB to alleviate spurious correlations. As shown in Figure 2, our model proceeds in four steps. **1)** The IB removes unimodal noise irrelevant to prediction, producing compact intermediate representations. **2)** These representations are stacked across modalities, and a self-attention module captures inter-modal and token-level dependencies to generate instrumental variables, which provide auxiliary signals for disentanglement. **3)** A learnable mask generator partitions the fused representation into causal and shortcut subrepresentations and performs disentanglement. **4)** Finally, a backdoor adjustment strategy randomly recombines causal and shortcut features to reduce their correlation. Furthermore, we provide a rigorous theoretical analysis in Subsection 4.4.

### 4.1 INFORMATION BOTTLENECK FILTERING

Given multimodal inputs $X = \{X_1, X_2, ..., X_M\}$, where $M$ represents the number of modalities. We first apply the IB principle to each unimodal input prior to fusion, with the goal of learning compact yet discriminative representations while filtering out noise irrelevant to prediction. Specifically, IB compresses the input state $X_i$ into a latent state $Z_i$, thereby minimizing redundant information while preserving its relevance to the label $Y$. This trade-off can be formalized via MI as the following variational optimization problem:

$$\min_{p(z_i|x_i)} I(X_i; Z_i) - \beta I(Z_i; Y) \tag{1}$$

where $I(\cdot; \cdot)$ denotes MI, and $\beta$ is a trade-off parameter that balances compression against predictive sufficiency. To optimize Eq. 1, for each modality $i$, we adopt a variational autoencoder ($VAE_i$) to map the input $X_i$ into the mean $\mu_i$ and variance $\sigma_i$ of a Gaussian distribution:

$$\mu_i, \sigma_i = VAE_i(x_i; \theta_{VAE_i}) \tag{2}$$

where $\theta_{VAE_i}$ denotes the parameters of $VAE_i$. To improve training efficiency and enable gradient propagation, we leverage the reparameterization trick to obtain the latent vector $z_i$:

$$z_i = \mu_i + \sigma_i \times \varepsilon_i, \quad \varepsilon_i \sim \mathcal{N}(0, I) \tag{3}$$

Finally, Eq. 1 can be approximated by the following tractable objective:

$$\begin{aligned} I(X_i; Z_i) - \beta I(Z_i; Y) \approx \mathbb{E}_{p(x_i)} \, \mathrm{KL}\big(p_\theta(z_i|x_i) \,\|\, q(z_i)\big) \\ - \beta \cdot \mathbb{E}_{p(x_i, y)} \mathbb{E}_{p_\theta(z_i|x_i)} \big[ \log q_\psi(y|z_i) \big] \end{aligned} \tag{4}$$

where $\mathrm{KL}\,(p_\theta(z_i|x_i), |, q(z_i))$ denotes the Kullback–Leibler (KL) divergence between the approximate posterior distribution $p_\theta(z_i|x_i)$ and the prior distribution $q(z_i)$. By minimizing this objective, the model learns compact latent representations $Z_i$ that serve as an information bottleneck between $X_i$ and $Y$, retaining task-relevant information while filtering out irrelevant noise. Further analysis is provided in the Appendix A.1.

## 4.2 MODELING CAUSAL INSTRUMENTAL VARIABLES

Statistically, causal substructures are typically determined by the global attributes of multimodal inputs rather than by any single modality or local features (Fan et al., 2022). Therefore, accurately extracting causal substructures requires modeling both the complex interactions across modalities and the contextual dependencies within each modality. To this end, we introduce an instrumental variable (Baiocchi et al., 2014; Wang et al., 2024) $V$ to help the model capture causal features $Z_c$ while mitigating the influence of shortcut factors $Z_s$.

Let the compressed representations obtained via the IB be stacked across modalities, forming a tensor $Z \in \mathbb{R}^{M \times L \times d}$, where $M$ denotes the number of modalities, $L$ the sequence length, and $d$ the feature dimension. We explicitly model the inter-modal and token-level dependencies of the instrumental variable using a self-attention mechanism:

$$\hat{V}_i = \sum_{j=1}^{M \cdot L} \frac{\exp(s_{ij})}{\sum_{n=1}^{M \cdot L} \exp(s_{in})} \, v_j, \quad s_{ij} = \frac{q_i^\top k_j}{\sqrt{d}}, \tag{5}$$

where $q_i = z_i W_Q$, $k_j = z_j W_K$, and $v_j = z_j W_V$ denote the query, key, and value vectors for tokens $i$ and $j$, respectively, with $W_Q$, $W_K$, and $W_V$ as the corresponding projection matrices. Here, $z_i$ denotes the representation of the $i$-th token, flattened from $Z$ across all modalities and positions. The resulting $\hat{V}$ is subsequently reshaped along the modality dimension and aggregated to yield the final instrumental variable:

$$V = \Big[ \sum_{m=1}^{M} \hat{V}_{m,1}, \ \sum_{m=1}^{M} \hat{V}_{m,2}, \ \ldots, \ \sum_{m=1}^{M} \hat{V}_{m,L} \Big] \tag{6}$$

The instrumental variable $V \in \mathbb{R}^{L \times d}$ captures both inter-modal and token-level dependencies, serving as a crucial auxiliary signal for subsequent causal modeling and enabling effective separation of causal features $Z_c$ from shortcut factors $Z_s$. A formal proof is provided in Subsection 4.4.

## 4.3 LEARNING DISENTANGLED CAUSAL AND SHORTCUT SUBREPRESENTATIONS

Given the intermediate representations filtered by the IB, $Z_1, Z_2, \ldots, Z_M$, we first concatenate them to obtain a fused representation $Z_m$. We then leverage a generative probabilistic model to decompose $Z_m$ into causal and shortcut subrepresentations. Concretely, a multilayer perceptron (MLP) estimates the probability $c_{ij}$ that each element of $Z_m$ belongs to the causal subrepresentation, and then maps it to the range $(0, 1)$ via a sigmoid function $\sigma$:

$$c_{ij} = \sigma(\text{MLP}(Z_m)), \quad Z_m = Fusion(Concat(Z_1, Z_2, ..., Z_M)) \tag{7}$$

The probability of belonging to the shortcut subrepresentation is then $b_{ij} = 1 - c_{ij}$. Using these probabilities, we construct the causal and shortcut masks $M_c = [c_{ij}]$ and $M_s = [b_{ij}]$, and decompose the multimodal representation $Z_m$ into its causal and shortcut subrepresentations:

$$Z_c = M_c \odot Z_m, \quad Z_s = M_s \odot Z_m \tag{8}$$

Given $Z_c$ and $Z_s$, how can we ensure that they correspond to the causal subrepresentation and the shortcut subrepresentation, respectively? Our goal is to guarantee that each captures the intended semantics. For $Z_c$, on one hand, we encourage its representation to align with the instrumental variable $V$ while reducing its correlation with $Z_s$; on the other hand, we leverage the task supervision signal to ensure that predictions based on $Z_c$ faithfully reflect the true labels. The corresponding loss functions are defined as follows:

$$\mathcal{L}^{Z_c \to V} = \|Z_c - V\|^2 \tag{9}$$

$$\mathcal{L}_{caus} = -\frac{1}{N} \sum_{n=1}^{N} \log p(\hat{y}^n \mid z_c^n) \tag{10}$$

where $p(\hat{y} \mid z_c)$ denotes the prediction distribution based on $Z_c$. For classification tasks, we define $\mathcal{L}_{caus} = CE(\hat{y}, y)$ using the cross-entropy loss, while for regression tasks, the mean squared error $MSE(\hat{y}, y)$ is employed. To suppress task-related information in the shortcut subrepresentation

$Z_s$, we enforce that the prediction distribution based on $Z_s$ approximates an uninformative uniform prior. This ensures that $Z_s$ does not provide a reliable pathway for solving the task:

$$\mathcal{L}_{unif} = \frac{1}{N} \sum_{n=1}^{N} \mathrm{KL}\big(p(\hat{y}^n \mid z_s^n) \,\big\|\, y_{unif}\big) \tag{11}$$

where $\mathrm{KL}(\cdot\|\cdot)$ denotes the KL divergence, and $y_{unif}$ represents the uniform prior: for classification tasks, $y_{unif} = (1/C, \ldots, 1/C)$ over $C$ classes; for regression tasks, it can be modeled as a Gaussian distribution with zero mean and variance matched to the dynamic range of the target values.

To further reduce the correlation between $Z_c$ and $Z_s$ and improve robustness under distribution shifts, we adopt an intervention strategy based on the backdoor adjustment (Pearl, 2009; Sui et al., 2022). Specifically, the causal subrepresentation is randomly combined with shortcut subrepresentations from other samples, so that the model is encouraged to rely on the causal information for accurate predictions, regardless of the spurious information. The intervention loss is defined as:

$$\mathcal{L}_{intv} = -\frac{1}{N \cdot |\hat{\mathcal{S}}|} \sum_{n=1}^{N} \sum_{z_s^{(k)} \in \hat{\mathcal{S}}} \log p\big(\hat{y}^n \mid z^{'}\big), \quad z^{'} = z_c^n + z_s^{(k)} \tag{12}$$

where $N$ is the number of samples, $\hat{\mathcal{S}}$ denotes the set of shortcut subrepresentations sampled from other instances, with cardinality $|\hat{\mathcal{S}}|$. The final training objective is the weighted sum of all losses:

$$\mathcal{L} = \mathcal{L}_{caus} + \lambda_1(\mathcal{L}^{Z_c \to V} + \mathcal{L}_{unif}) + \lambda_2\mathcal{L}_{intv} \tag{13}$$

where the hyperparameters $\lambda_1$ and $\lambda_2$ control the relative weights of the disentanglement losses and the causal intervention loss.

## 4.4 THEORETICAL ANALYSIS

In this subsection, we provide a theoretical analysis of the above procedure. Assume the attention weight for token $i$ on token $j$ is

$$\alpha_{ij} = \frac{\exp(s_{ij})}{\sum_m \exp(s_{im})} \tag{14}$$

Its derivative with respect to the $s_{ij}$ is

$$\frac{\partial \alpha_{im}}{\partial s_{ij}} = \alpha_{im}(\delta_{mj} - \alpha_{ij}) \tag{15}$$

where $\delta_{mj}$ is the Kronecker delta. Defining $\hat{V}_i = \sum_m \alpha_{im} v_m$, we obtain

$$\frac{\partial \hat{V}_i}{\partial s_{ij}} = \alpha_{ij}(v_j - \hat{V}_i) \tag{16}$$

indicating that adjusting $s_{ij}$ moves $\hat{V}i$ toward $v_j$ by strength $\alpha_{ij}$. Let $z_i$ be the input of token $i$, with $q_i = z_i W_Q$, $k_j = z_j W_K$, and $v_j = z_j W_V$. The gradient of the loss with respect to $s_{ij}$ is

$$\frac{\partial \mathcal{L}}{\partial s_{ij}} = \alpha_{ij}\Big\langle \frac{\partial \mathcal{L}}{\partial \hat{V}_i}, v_j - \hat{V}_i \Big\rangle \tag{17}$$

showing that scores linked to causal features $Z_c$ (reducing loss) are reinforced, while those aligned with shortcut features $Z_s$ are suppressed. For dot-product attention $s_{ij} = \frac{1}{\sqrt{d}} q_i^\top k_j$, we have

$$\frac{\partial s_{ij}}{\partial W_Q} = \frac{z_i k_j^\top}{\sqrt{d}}, \quad \frac{\partial s_{ij}}{\partial W_K} = \frac{z_j q_i^\top}{\sqrt{d}}, \quad \frac{\partial s_{ij}}{\partial W_V} = 0 \tag{18}$$

so $W_Q$ and $W_K$ directly shape attention, while $W_V$ influences the output via $v_j$. To regularize attention, we introduce two constraints. First, minimizing

$$\mathrm{KL}(p(\hat{y}^n \mid z_s^n) \,\|\, y_{unif}) = \log K - H(p(\hat{y}^n \mid z_s^n)) \tag{19}$$

is equivalent to maximizing the entropy, which prevents the model from relying on $Z_s$, where $K$ denotes the number of classes. Second, minimizing $\|Z_c - V\|^2$ yields gradients

$$\frac{\partial}{\partial V}\|Z_c - V\|^2 = -2(Z_c - V), \quad \frac{\partial}{\partial Z_c}\|Z_c - V\|^2 = 2(Z_c - V) \qquad (20)$$

which align the instrumental variable $V$ with the causal subrepresentation $Z_c$.

In summary, let $Z$ denote the final learned representation that integrates causal information. Attention optimization combined with KL and MSE regularization ensures that: **i)** weights on causal features $Z_c$ are strengthened, while those on $Z_s$ are suppressed; **ii)** $V$ is sensitive to $Z_c$ but not to $Z_s$, guaranteeing $P(Z \mid V) \neq P(Z)$ and $P(Z_s \mid V) = P(Z_s)$; **iii)** $Z$ blocks shortcut paths, ensuring $P(Y \mid Z, V) = P(Y \mid Z_c)$. Therefore, incorporating inter-modal and token-level self-attention with appropriate regularization enables the extraction of robust causal subrepresentations in multimodal learning. The detailed proof is provided in the Appendix A.2.

## 5 EXPERIMENTS

In this section, we evaluate CaMIB on four widely used multimodal datasets: the Multimodal Sentiment Analysis (MSA) datasets CMU-MOSI (Zadeh et al., 2016) and CMU-MOSEI (Zadeh et al., 2018), the Multimodal Humor Detection (MHD) dataset UR-FUNNY (Hasan et al., 2019), and the Multimodal Sarcasm Detection (MSD) dataset MUStARD (Castro et al., 2019). To further assess model generalizability under distribution shifts, we also conduct experiments on the OOD variant of CMU-MOSI, with data splitting following (Sun et al., 2022). For brevity, dataset information, evaluation metrics, baselines, implementation details, and additional results are in the Appendix.

### 5.1 MULTIMODAL SENTIMENT ANALYSIS

Table 1: Comparison on the CMU-MOSI and CMU-MOSEI datasets. Acc2 and F1 scores are reported in two configurations: negative/non-negative (including zero) and negative/positive (excluding zero). $d$ indicates results from our reproduced experiments, which also use the DeBERTa pre-trained model. The best and second results are highlighted with bold and underline, respectively.

| Model | CMU-MOSI | | | | | CMU-MOSEI | | | | |
|---|---|---|---|---|---|---|---|---|---|---|
| | Acc7↑ | Acc2↑ | F1↑ | MAE↓ | Corr↑ | Acc7↑ | Acc2↑ | F1↑ | MAE↓ | Corr↑ |
| Self-MM | - | 84.0/86.0 | 84.4/86.0 | 0.713 | 0.798 | - | 82.8/85.2 | 82.5/85.3 | 0.530 | 0.765 |
| MMIM | 46.7 | 84.1/86.1 | 84.0/86.0 | 0.700 | 0.800 | 54.2 | 82.2/86.0 | 82.7/86.0 | 0.526 | 0.772 |
| HyCon | 46.6 | -/85.2 | -/85.1 | 0.713 | 0.790 | 52.8 | -/85.4 | -/85.6 | 0.601 | 0.776 |
| ConFEDE | 42.3 | 84.2/85.5 | 84.1/85.5 | 0.742 | 0.784 | **54.9** | 81.7/85.8 | 82.2/85.8 | 0.522 | 0.780 |
| KuDA | 47.1 | 84.4/86.4 | 84.5/86.5 | 0.705 | 0.795 | 52.9 | 83.3/86.5 | 83.0/86.6 | 0.529 | 0.776 |
| DLF | 47.1 | -/85.1 | -/85.0 | 0.731 | 0.781 | 53.9 | -/85.4 | -/85.3 | 0.536 | 0.764 |
| DEVA | 46.3 | 84.4/86.3 | 84.5/86.3 | 0.730 | 0.787 | 52.3 | 83.3/86.1 | 82.9/86.2 | 0.541 | 0.769 |
| E-MIB$_d$ | 47.6 | 86.3/87.6 | 86.2/87.6 | 0.646 | 0.845 | 53.1 | 83.0/86.5 | 83.4/86.5 | 0.528 | 0.778 |
| L-MIB$_d$ | **48.0** | 86.3/88.2 | 86.2/88.2 | 0.636 | 0.848 | 53.1 | 84.0/86.8 | 84.3/86.8 | 0.542 | 0.777 |
| C-MIB$_d$ | 47.6 | 85.4/87.2 | 85.3/87.2 | 0.650 | 0.840 | 53.8 | 83.7/86.6 | 84.1/86.6 | 0.526 | 0.779 |
| ITHP$_d$ | 46.3 | 86.1/88.2 | 86.0/88.2 | 0.654 | 0.844 | 51.6 | 82.3/86.2 | 82.9/86.3 | 0.556 | 0.781 |
| CaMIB | **48.0** | **88.2/89.8** | **88.1/89.8** | **0.616** | **0.857** | 53.5 | **85.3/87.3** | **85.4/87.2** | **0.517** | **0.788** |

We evaluated CaMIB on two widely used MSA datasets and compared it with several competitive baselines. As shown in Table 1, CaMIB outperforms most baselines across multiple evaluation metrics and demonstrates consistent advantages on both CMU-MOSI and CMU-MOSEI. Specifically, on CMU-MOSI, CaMIB achieves an Acc7 score of 48.0%, tying with L-MIB (Mai et al., 2023c) for the highest among all baselines, and surpassing ITHP (Xiao et al., 2024)—which also employs DeBERTa (He et al., 2020) as the language encoder—by 1.7%. CaMIB additionally attains the highest Acc2 and F1 scores among all baselines, outperforming the second-best methods by 1.6%–1.9%. Moreover, CaMIB substantially surpasses existing approaches in MAE and Acc2 (including zero), highlighting its strong capability in predicting neutral sentiment. On CMU-MOSEI, CaMIB achieves an Acc7 score slightly lower (by 0.3%) than C-MIB (Mai et al., 2023c), which also uses DeBERTa, but still outperforms all other models based on the same language network. Furthermore,

CaMIB exceeds all baselines on the remaining metrics and improves Acc2 (including zero) by 1.1%–1.3% over the second-best method. Overall, considering results on both datasets, CaMIB achieves state-of-the-art performance in MSA tasks. Given that current top-performing methods have already surpassed human-level performance (Xiao et al., 2024), these improvements are substantial.

In addition, we compared CaMIB with MIB variants at different fusion stages: early fusion (E-MIB), late fusion (L-MIB), and the combined framework (C-MIB). Experimental results show that CaMIB consistently outperforms these baselines on both datasets. Unlike traditional IB methods that overemphasize maximizing mutual information while neglecting spurious correlations, CaMIB leverages disentangled causal learning to effectively mitigate bias and enhance generalization. Notably, although CaMIB does not achieve the highest Acc7 on CMU-MOSEI, its performance can be further improved by tuning the strength of disentanglement and causal intervention. As shown in Appendix B.1, under various parameter settings, CaMIB achieves Acc7 scores exceeding 54% on CMU-MOSEI in most cases, demonstrating the model's robustness and adaptability.

## 5.2 MULTIMODAL HUMOR AND SARCASM DETECTION

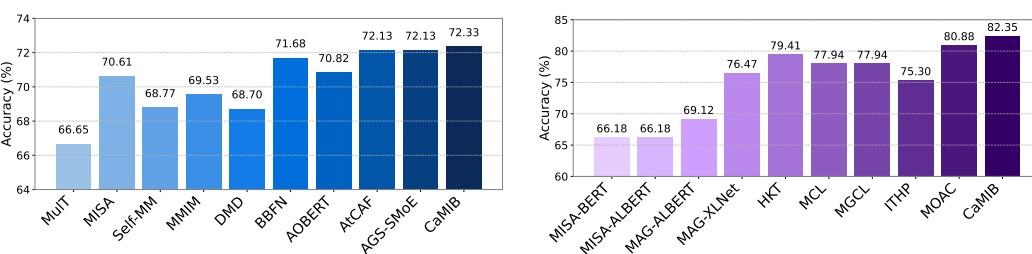

Figure 3: Comparison on the UR-FUNNY dataset (left) and the MUStARD dataset (right).

To further evaluate the generalizability of CaMIB across different MLU tasks, we conducted experiments on the UR-FUNNY and MUStARD datasets for the MHD and MSD tasks, respectively. As illustrated in Figure 3, for the MHD task, CaMIB outperforms the latest state-of-the-art methods, including AtCAF (Huang et al., 2025) and AGS-SMoE (Chen et al., 2025). For the MSD task, CaMIB achieves substantial improvements over all baseline models, surpassing the second-best method MOAC (Mai et al., 2025a) by 1.47% in accuracy. Notably, compared with the IB–based method ITHP (Xiao et al., 2024), CaMIB attains a remarkable gain of 7.05%. Overall, CaMIB establishes new state-of-the-art performance on both MHD and MSD tasks, demonstrating its effectiveness and generalizability across MLU tasks.

## 5.3 OUT-OF-DISTRIBUTION EXPERIMENTS

Table 2 presents the performance comparison between CaMIB and other methods under OOD test settings. Several observations can be drawn: **i)** Performance under OOD testing is lower than on the standard dataset for all methods, confirming that spurious correlations indeed undermine generalization ability; **ii)** Under the OOD settings, CaMIB significantly outperforms the baseline based on conventional multimodal fusion techniques, with ACC2 and F1 improvements over ITHP (Xiao et al., 2024) from 2.1%/1.6% to 3.3%/3.2%, demonstrating the effectiveness of our causal debiasing strategy in enhancing generalization; **iii)** Compared with causal-based baselines such as CLUE (Sun et al., 2022), MulDeF (Huan et al., 2024), GEAR (Sun et al., 2023), and the recent MMCI (Jiang et al., 2025),

Table 2: Comparison on the OOD version of the CMU-MOSI dataset.

| Model | CMU-MOSI (OOD) | | |
|---|---|---|---|
| | Acc7↑ | Acc2↑ | F1↑ |
| Self-MM | 40.2 | 76.7/78.1 | 76.7/78.1 |
| ITHP | 43.0 | 79.5/81.2 | 79.5/81.3 |
| CLUE | 41.8 | 78.8/79.9 | 78.8/79.9 |
| GEAR | - | 80.5/82.1 | 80.4/82.1 |
| MulDeF | 42.9 | 79.8/81.4 | 79.9/81.5 |
| MMCI | 44.5 | 81.2/83.3 | 81.2/83.3 |
| CaMIB | **45.0** | **82.8/84.4** | **82.7/84.4** |

CaMIB achieves superior performance across all metrics, highlighting the advantage of performing causal intervention directly on the fused representation without relying on predefined bias types.

## 5.4 ABLATION STUDIES

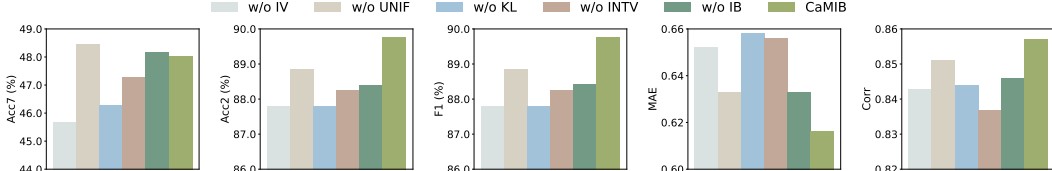

Figure 4: Ablation experiments on the CMU-MOSI dataset.

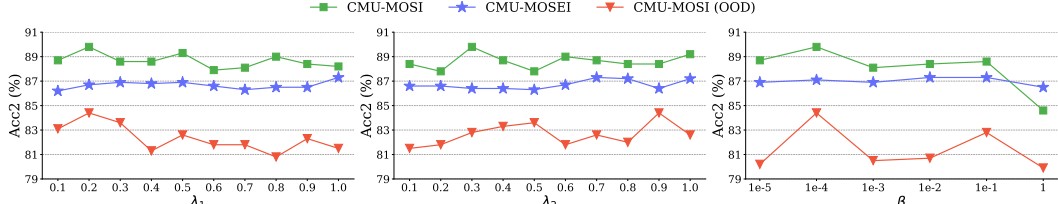

Figure 5: Sensitivity analysis of parameters $\lambda_1$, $\lambda_2$, and $\beta$.

In this Subsection, we present ablation experiments to assess the contribution of each component in CaMIB: **1) Importance of the instrumental variable constraint.** In this setting, we remove the instrumental variable constraint on $Z_c$ ("w/o IV"). As shown in Figure 4, model performance drops significantly and becomes nearly the worst among all variants. This highlights the crucial role of leveraging the self-attention mechanism to capture cross-modal and token-level dependencies, thereby extracting global causal features. **2) Importance of suppressing task-relevant information in $Z_s$.** In the "w/o UNIF" setting, removing $\mathcal{L}_{unif}$ results in the smallest performance drop, as the shortcut representation initially contains limited mutual information with the labels, which restricts the space for partial disentanglement. Further evidence comes from another experiment: replacing the prediction loss on the shortcut representation with MSE ("w/o KL") reduces CaMIB to a model with only instrumental variable constraints and insufficient disentanglement. As shown in Figure 4, this results in a sharp decline across all metrics, highlighting the necessity of suppressing task-relevant information in $Z_s$. **3) Importance of causal intervention.** In the "w/o INTV" configuration, we set $\lambda_2 = 0$ to disable random recombination of the causal and shortcut subrepresentation. This leads to performance degradation across multiple metrics, with a larger drop than in "w/o UNIF," further emphasizing the critical role of causal intervention. **4) Importance of information bottleneck filtering.** In this setting, we remove information bottleneck filtering on unimodal features ("w/o IB"). The performance drop suggests that eliminating irrelevant noise and obtaining compact representations benefits the model. Even without IB, the model still outperforms L-MIB, showing that causal methods alone can surpass purely information-bottleneck-based approaches, further supporting the effectiveness of our approach. **5) Sensitivity analysis of parameters $\lambda_1$, $\lambda_2$, and $\beta$.** According to Equation 13, $\lambda_1$ controls disentangling strength between causal and shortcut features, $\lambda_2$ governs causal intervention intensity, and $\beta$ balances compression and prediction objectives. Experiments (details in Appendix B.1) show (Figure 5): **i)** Under the OOD settings, performance is more sensitive to $\lambda_1$ and $\lambda_2$, indicating these parameters should be chosen carefully; **ii)** $\beta$ should not be too large or too small—too large degrades performance, while too small leaves residual noise in the filtered unimodal information, potentially affecting downstream processing.

## 6 CONCLUSION

We observe that most existing works predominantly follow the "learning to attend" paradigm, which degrades OOD generalization by conflating statistical shortcuts with genuine causal features. In this work, we propose a Causal Multimodal Information Bottleneck (CaMIB) model that effectively captures global causal features while suppressing irrelevant noise. Extensive experiments on multiple MLU tasks and OOD test sets demonstrate that CaMIB achieves superior performance and robustness. Theoretical and empirical analyses further validate the interpretability and sound causal principles of our approach, providing a new perspective to the MLU community.

## REPRODUCIBILITY STATEMENT

We have taken extensive measures to ensure the reproducibility of our work. The complete and executable source code is provided in the supplementary materials, along with a detailed README containing step-by-step instructions for all experiments. All datasets used are publicly available, with detailed information and preprocessing procedures provided in Appendix C. Theoretical analyses, including all assumptions and full proofs, are presented in Appendix A. Hyperparameter settings and training configurations are fully specified in Appendix D to facilitate reproducibility. Together, these resources enable full replication and verification of our results.

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

# Appendix

CONTENTS

## THE USE OF LARGE LANGUAGE MODELS (LLMs)

In this work, large language models (LLMs) were used solely as a general-purpose tool to assist in language polishing and improving the clarity and fluency of the manuscript. All scientific content, experimental design, results, and conclusions are entirely the responsibility of the authors. No LLM was involved in ideation, analysis, or interpretation of the research, and all content generated by LLMs was carefully reviewed and verified by the authors.

## A  THEORETICAL ANALYSIS

### A.1  DERIVATIONS OF THE INFORMATION BOTTLENECK FILTERING

In this subsection, we provide a detailed derivation from the original IB objective in Eq. 1 to the trainable variational form in Eq. 4. Recall that, given a unimodal input $X$, its compressed representation $Z$, and label $Y$, the IB objective is defined as

$$\min_{p(z|x)} \; I(X;Z) - \beta \, I(Z;Y) \tag{21}$$

where $\beta$ is a trade-off parameter that balances the two mutual information terms. Direct computation of the mutual information between $X/Z/Y$ is generally intractable. Therefore, we aim to obtain an upper bound for $I(X;Z) - \beta \, I(Z;Y)$ and convert the minimization problem into an evidence bound optimization problem.

We first rewrite the two mutual information terms in forms that are amenable to approximation and computation, and introduce trainable variational distributions to obtain a solvable objective.

**Step 1: Express mutual information in full form**

$$
\begin{aligned}
I(X;Z) &= \iint p(x,z) \log \frac{p(z|x)}{p(z)} \, dz \, dx \\
&= \int p(x) \left[ \int p(z|x) \log \frac{p(z|x)}{p(z)} \, dz \right] dx \\
&= \mathbb{E}_{p(x)} \left[ \int p(z|x) \log \frac{p(z|x)}{p(z)} \, dz \right] \\
&= \mathbb{E}_{p(x)} \left[ \mathrm{KL}\big(p(z|x) \,\|\, p(z)\big) \right]
\end{aligned}
\tag{22}
$$

Similarly, $I(Z;Y)$ can be written as

$$
\begin{aligned}
I(Z;Y) &= \iint p(z,y) \log \frac{p(y|z)}{p(y)} \, dz \, dy \\
&= \iint p(z,y) \log p(y|z) \, dz \, dy - \iint p(z,y) \log p(y) \, dz \, dy \\
&= \mathbb{E}_{p(z,y)} \big[ \log p(y|z) \big] - \mathbb{E}_{p(y)} \log p(y) \\
&= \mathbb{E}_{p(x,y)} \mathbb{E}_{p(z|x)} \big[ \log p(y|z) \big] - H(Y)
\end{aligned}
\tag{23}
$$

where $H(Y) = -\mathbb{E}_{p(y)} \log p(y)$ is constant with respect to encoder parameters.

**Step 2: Variational upper bound for $I(X;Z)$**

$$
\begin{aligned}
\mathbb{E}_{p(x)} \mathrm{KL}\big(p_\theta(z|x) \,\|\, q(z)\big) &= \mathbb{E}_{p(x)} \int p_\theta(z|x) \log \frac{p_\theta(z|x)}{q(z)} dz \\
&= \mathbb{E}_{p(x)} \int p_\theta(z|x) \left[ \log \frac{p_\theta(z|x)}{p(z)} + \log \frac{p(z)}{q(z)} \right] dz \\
&= \mathbb{E}_{p(x)} \int p_\theta(z|x) \log \frac{p_\theta(z|x)}{p(z)} dz + \mathbb{E}_{p(x)} \int p_\theta(z|x) \log \frac{p(z)}{q(z)} dz \\
&= \mathbb{E}_{p(x)} \mathrm{KL}\big(p_\theta(z|x) \,\|\, p(z)\big) + \int p(z) \log \frac{p(z)}{q(z)} dz \\
&= \mathbb{E}_{p(x)} \mathrm{KL}\big(p_\theta(z|x) \,\|\, p(z)\big) + \mathrm{KL}\big(p(z) \,\|\, q(z)\big)
\end{aligned}
\tag{24}
$$

where $p(z) = \int p(x)p_\theta(z|x)\,dx$. Then we have

$$I(X;Z) = \mathbb{E}_{p(x)}\mathrm{KL}\big(p_\theta(z|x)\,\|\,q(z)\big) - \mathrm{KL}\big(p(z)\,\|\,q(z)\big) \leq \mathbb{E}_{p(x)}\mathrm{KL}\big(p_\theta(z|x)\,\|\,q(z)\big) \qquad (25)$$

**Step 3: Variational lower bound for $I(Z;Y)$.** Introduce a variational distribution $q_\psi(y|z)$ to approximate the true posterior $p(y|z)$. By the non-negativity of the KL divergence:

$$\mathrm{KL}\big(p(y|z)\,\|\,q_\psi(y|z)\big) \geq 0$$

$$\Rightarrow \mathbb{E}_{p(y|z)} \log \frac{p(y|z)}{q_\psi(y|z)} \geq 0 \qquad (26)$$

$$\Rightarrow \mathbb{E}_{p(y|z)} \log p(y|z) \geq \mathbb{E}_{p(y|z)} \log q_\psi(y|z)$$

and taking the expectation over $p(z)$ gives:

$$\mathbb{E}_{p(z)}\mathbb{E}_{p(y|z)}\big[\log p(y|z)\big] \geq \mathbb{E}_{p(z)}\mathbb{E}_{p(y|z)}\big[\log q_\psi(y|z)\big] \qquad (27)$$

Using the joint distribution $p(z,y) = p(z)p(y|z)$, the inequality can be equivalently written as:

$$\mathbb{E}_{p(z,y)}\big[\log p(y|z)\big] \geq \mathbb{E}_{p(z,y)}\big[\log q_\psi(y|z)\big] \qquad (28)$$

Further, incorporating the encoder-induced distribution $p_\theta(z|x)$ yields:

$$\mathbb{E}_{p(x,y)}\mathbb{E}_{p_\theta(z|x)}\big[\log p(y|z)\big] \geq \mathbb{E}_{p(x,y)}\mathbb{E}_{p_\theta(z|x)}\big[\log q_\psi(y|z)\big] \qquad (29)$$

Therefore, a variational lower bound for $I(Z;Y)$ can be expressed as:

$$I(Z;Y) \geq \mathbb{E}_{p(x,y)}\mathbb{E}_{p_\theta(z|x)} \log q_\psi(y|z) - H(Y). \qquad (30)$$

where $H(Y) = -\mathbb{E}_{p(y)} \log p(y)$ is independent of the model parameters $\theta$ and $\psi$, and can thus be treated as a constant during optimization.

**Step 4: Combine bounds into the IB objective**

$$\begin{aligned} I(X;Z) - \beta\,I(Z;Y) &\leq \mathbb{E}_{p(x)}\mathrm{KL}\big(p_\theta(z|x)\,\|\,q(z)\big) - \beta\,\mathbb{E}_{p(x,y)}\mathbb{E}_{p_\theta(z|x)}\big[\log q_\psi(y|z)\big] \\ &\approx \mathbb{E}_{p(x)}\mathrm{KL}\big(p_\theta(z|x)\,\|\,q(z)\big) - \beta\,\mathbb{E}_{p(x,y)}\mathbb{E}_{p_\theta(z|x)}\big[\log q_\psi(y|z)\big] \end{aligned} \qquad (31)$$

In practice, the encoder $p_\theta(z|x)$ is modeled as a diagonal Gaussian:

$$p_\theta(z|x) = \mathcal{N}\big(z;\,\mu_\theta(x),\,diag(\sigma_\theta^2(x))\big) \qquad (32)$$

and the reparameterization trick is applied:

$$z = \mu_\theta(x) + \sigma_\theta(x) \odot \varepsilon, \qquad \varepsilon \sim \mathcal{N}(0, I) \qquad (33)$$

With a standard normal prior $q(z) = \mathcal{N}(0, I)$, the KL term has an analytical solution:

$$\mathrm{KL}\big(\mathcal{N}(\mu, \mathrm{diag}(\sigma^2))\,\|\,\mathcal{N}(0, I)\big) = \tfrac{1}{2} \sum_{j=1}^{d} \Big(\sigma_j^2 + \mu_j^2 - 1 - \log \sigma_j^2\Big) \qquad (34)$$

For the second term, a Monte Carlo approximation is used:

$$\mathbb{E}_{p_\theta(z|x)}[\log q_\psi(y|z)] \approx \frac{1}{L} \sum_{l=1}^{L} \log q_\psi\big(y \mid z^{(l)}\big), \qquad z^{(l)} = \mu_\theta(x) + \sigma_\theta(x) \odot \varepsilon^{(l)} \qquad (35)$$

Finally, the per-sample approximate loss is

$$\mathcal{L}(x, y) \approx \mathrm{KL}\big(\mathcal{N}(\mu, \sigma^2)\,\|\,\mathcal{N}(0, I)\big) - \beta \cdot \frac{1}{L} \sum_{l=1}^{L} \log q_\psi\big(y \mid z^{(l)}\big) \qquad (36)$$

Averaging over a minibatch and performing stochastic gradient descent on $(\theta, \psi)$ yields the trainable IB optimization procedure.

## A.2 DETAILED ANALYSIS OF CaMIB

In this subsection, we provide detailed derivations for the theoretical results presented in Section 4.4. Starting from the attention weight definition:

$$\alpha_{ij} = \frac{\exp(s_{ij})}{\sum_m \exp(s_{im})} \tag{37}$$

We compute the partial derivative $\frac{\partial \alpha_{im}}{\partial s_{ij}}$ using the quotient rule. Consider two cases:

**Case 1:** $m = j$

$$\begin{aligned}
\frac{\partial \alpha_{ij}}{\partial s_{ij}} &= \frac{\exp(s_{ij}) \sum_m \exp(s_{im}) - \exp(s_{ij}) \exp(s_{ij})}{\left(\sum_m \exp(s_{im})\right)^2} \\
&= \frac{\exp(s_{ij})}{\sum_m \exp(s_{im})} - \frac{\exp(s_{ij})}{\sum_m \exp(s_{im})} \cdot \frac{\exp(s_{ij})}{\sum_m \exp(s_{im})} \\
&= \alpha_{ij} - \alpha_{ij}^2 = \alpha_{ij}(1 - \alpha_{ij})
\end{aligned} \tag{38}$$

**Case 2:** $m \neq j$

$$\begin{aligned}
\frac{\partial \alpha_{im}}{\partial s_{ij}} &= \frac{0 \cdot \sum_k \exp(s_{ik}) - \exp(s_{im}) \exp(s_{ij})}{\left(\sum_k \exp(s_{ik})\right)^2} \\
&= -\frac{\exp(s_{im})}{\sum_k \exp(s_{ik})} \cdot \frac{\exp(s_{ij})}{\sum_k \exp(s_{ik})} \\
&= -\alpha_{im}\alpha_{ij}
\end{aligned} \tag{39}$$

Combining both cases using the Kronecker delta $\delta_{mj}$ (which equals 1 when $m = j$ and 0 otherwise):

$$\frac{\partial \alpha_{im}}{\partial s_{ij}} = \alpha_{im}(\delta_{mj} - \alpha_{ij}) \tag{40}$$

Given the attention-weighted value vector $\hat{V}_i = \sum_m \alpha_{im} v_m$, the derivative with respect to $s_{ij}$ is:

$$\begin{aligned}
\frac{\partial \hat{V}_i}{\partial s_{ij}} &= \sum_m \frac{\partial \alpha_{im}}{\partial s_{ij}} v_m \\
&= \sum_m \alpha_{im}(\delta_{mj} - \alpha_{ij}) v_m \\
&= \sum_m \alpha_{im} \delta_{mj} v_m - \alpha_{ij} \sum_m \alpha_{im} v_m \\
&= \alpha_{ij} v_j - \alpha_{ij} \hat{V}_i \\
&= \alpha_{ij}(v_j - \hat{V}_i)
\end{aligned} \tag{41}$$

which shows that adjusting $s_{ij}$ moves $\hat{V}_i$ toward $v_j$ with strength proportional to $\alpha_{ij}$.

Using the chain rule, the gradient of the loss with respect to $s_{ij}$ is:

$$\begin{aligned}
\frac{\partial \mathcal{L}}{\partial s_{ij}} &= \left\langle \frac{\partial \mathcal{L}}{\partial \hat{V}_i}, \frac{\partial \hat{V}_i}{\partial s_{ij}} \right\rangle \\
&= \left\langle \frac{\partial \mathcal{L}}{\partial \hat{V}_i}, \alpha_{ij}(v_j - \hat{V}_i) \right\rangle \\
&= \alpha_{ij} \left\langle \frac{\partial \mathcal{L}}{\partial \hat{V}_i}, v_j - \hat{V}_i \right\rangle
\end{aligned} \tag{42}$$

which demonstrates that attention scores associated with causal features $Z_c$ (which reduce the loss) are reinforced, while those aligned with shortcut features $Z_s$ are suppressed.

For dot-product attention with $s_{ij} = \frac{1}{\sqrt{d}} q_i^\top k_j$, where $q_i = z_i W_Q$ and $k_j = z_j W_K$:

Gradient with respect to $W_Q$:

$$
\begin{aligned}
\frac{\partial s_{ij}}{\partial W_Q} &= \frac{\partial}{\partial W_Q} \left( \frac{1}{\sqrt{d}} (z_i W_Q)^\top (z_j W_K) \right) \\
&= \frac{1}{\sqrt{d}} \frac{\partial}{\partial W_Q} \left( W_Q^\top z_i^\top z_j W_K \right) \\
&= \frac{1}{\sqrt{d}} z_i^\top z_j W_K \\
&= \frac{1}{\sqrt{d}} z_i^\top k_j = \frac{z_i k_j^\top}{\sqrt{d}}
\end{aligned}
\tag{43}
$$

Gradient with respect to $W_K$:

$$
\begin{aligned}
\frac{\partial s_{ij}}{\partial W_K} &= \frac{\partial}{\partial W_K} \left( \frac{1}{\sqrt{d}} (z_i W_Q)^\top (z_j W_K) \right) \\
&= \frac{1}{\sqrt{d}} \frac{\partial}{\partial W_K} \left( W_Q^\top z_i^\top z_j W_K \right) \\
&= \frac{1}{\sqrt{d}} W_Q^\top z_i^\top z_j \\
&= \frac{1}{\sqrt{d}} q_i^\top z_j = \frac{z_j q_i^\top}{\sqrt{d}}
\end{aligned}
\tag{44}
$$

Gradient with respect to $W_V$: Since $s_{ij}$ does not depend on $W_V$:

$$
\frac{\partial s_{ij}}{\partial W_V} = 0
\tag{45}
$$

which shows that $W_Q$ and $W_K$ directly shape the attention patterns, while $W_V$ influences the output through the value vectors $v_j$.

The KL divergence between the predicted distribution given shortcut features and a uniform distribution is:

$$
\begin{aligned}
\mathrm{KL}\left( p(\hat{y}^n \mid z_s^n) \,\|\, y_{unif} \right) &= \mathbb{E}_{p(\hat{y}^n|z_s^n)} \left[ \log \frac{p(\hat{y}^n \mid z_s^n)}{y_{unif}} \right] \\
&= \mathbb{E}_{p(\hat{y}^n|z_s^n)} \left[ \log p(\hat{y}^n \mid z_s^n) - \log y_{unif} \right] \\
&= \mathbb{E}_{p(\hat{y}^n|z_s^n)} \left[ \log p(\hat{y}^n \mid z_s^n) \right] - \log \frac{1}{K} \\
&= -H\left( p(\hat{y}^n \mid z_s^n) \right) + \log K
\end{aligned}
\tag{46}
$$

where $K$ is the number of classes and $y_{unif} = \frac{1}{K}$ is the uniform distribution. Therefore, minimizing the KL divergence is equivalent to maximizing the entropy $H\left( p(\hat{y}^n \mid z_s^n) \right)$, which prevents the model from relying on shortcut features $Z_s$.

For the mean squared error constraint $\|Z_c - V\|^2$:

Gradient with respect to $V$:

$$
\begin{aligned}
\frac{\partial}{\partial V} \|Z_c - V\|^2 &= \frac{\partial}{\partial V} \left[ (Z_c - V)^\top (Z_c - V) \right] \\
&= 2(Z_c - V)^\top (-1) = -2(Z_c - V)^\top
\end{aligned}
\tag{47}
$$

Gradient with respect to $Z_c$:

$$
\begin{aligned}
\frac{\partial}{\partial Z_c} \|Z_c - V\|^2 &= \frac{\partial}{\partial Z_c} \left[ (Z_c - V)^\top (Z_c - V) \right] \\
&= 2(Z_c - V)^\top
\end{aligned}
\tag{48}
$$

These gradients align the instrumental variable $V$ with the causal subrepresentation $Z_c$.

The combined optimization ensures three key properties: **i)** The gradient $\frac{\partial \mathcal{L}}{\partial s_{ij}} = \alpha_{ij} \left\langle \frac{\partial \mathcal{L}}{\partial \hat{V}_i}, v_j - \hat{V}_i \right\rangle$ strengthens weights on causal features $Z_c$ (loss-reducing) and suppresses weights on shortcut features $Z_s$. **ii)** The regularization ensures $V$ is sensitive to $Z_c$ but not to $Z_s$, guaranteeing

$P(Z \mid V) \neq P(Z)$ (relevance condition) and $P(Z_s \mid V) = P(Z_s)$ (exclusion restriction). **iii)** The learned representation $Z$ blocks shortcut paths, ensuring $P(Y \mid Z, V) = P(Y \mid Z_c)$, meaning $V$ provides no additional information about $Y$ given $Z$.

Therefore, the inter-modal and token-level self-attention mechanism with KL and MSE regularization enables robust extraction of causal subrepresentations in multimodal learning.

## B  ADDITIONAL EXPERIMENTAL RESULTS

### B.1  HYPERPARAMETER ANALYSIS

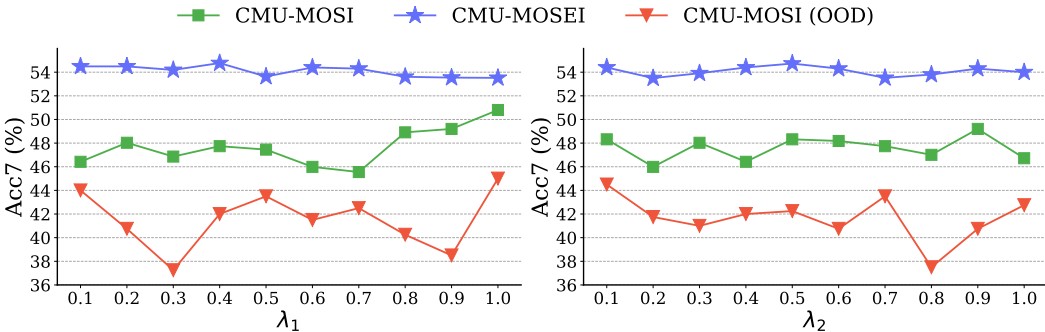

Figure 6: Sensitivity analysis of parameters $\lambda_1$ and $\lambda_2$ on the seven classification metrics (Acc7).

In this subsection, we further evaluate the impact of the hyperparameters $\lambda_1$ and $\lambda_2$ on Acc7, as a complement to the ablation studies presented in Section 5.4. Specifically, according to Equation 13, $\lambda_1$ controls the disentanglement strength between the causal and shortcut representations, while $\lambda_2$ governs the intensity of causal intervention. We adopt a grid search strategy with a step size of 0.1, fixing one coefficient at its optimal value reported in Table 6 and varying the other within the range $(0.1, 1)$. Experiments are conducted on the CMU-MOSI, CMU-MOSEI, and CMU-MOSI (OOD) datasets. Notably, for CMU-MOSI (OOD), the coefficient range is selected based on Acc2 to maintain consistency with the main text.

The results, shown in Figure 6, illustrate how Acc7 varies under different hyperparameter settings. CMU-MOSI (OOD) exhibits high sensitivity to both coefficients, with performance dropping significantly for certain values, e.g., $\lambda_1 = 0.3$ or $0.9$, and $\lambda_2 = 0.8$. In contrast, CMU-MOSEI is more robust to hyperparameter variations, which is reasonable given its substantially larger size compared to CMU-MOSI.

These findings suggest that careful tuning of the disentanglement and causal intervention strengths is particularly important in out-of-distribution scenarios, consistent with the trends observed in the Acc2 analysis in the main text. Smaller datasets tend to show greater fluctuations under parameter changes, while larger datasets remain relatively stable. It is also worth noting that most Acc7 results on CMU-MOSEI exceed 54%, outperforming the metrics reported in the main text, indicating that further performance gains for CaMIB can be achieved through appropriate hyperparameter adjustment. Overall, CaMIB demonstrates consistently strong and stable performance across most hyperparameter configurations, further validating its robustness.

### B.2  DISCUSSION OF THE PRE-TRAINED LANGUAGE MODEL

For our main task of MSA, following the state-of-the-art ITHP (Xiao et al., 2024), we adopt DeBERTa-v3-base (He et al., 2020) as the pre-trained language model (PLM). In this section, we evaluate and analyze the impact of different PLMs on overall performance, highlighting the benefits of our proposed CaMIB model.

As reported in Table 3, models using DeBERTa generally outperform their BERT-based counterparts on both CMU-MOSI and CMU-MOSEI datasets. For example, the DeBERTa-based MMIM achieves a higher correlation (0.829 vs. 0.800 on CMU-MOSI) and lower MAE (0.649 vs. 0.700)

Table 3: Performance comparison on the CMU-MOSI and CMU-MOSEI datasets. Models utilizing BERT and DeBERTa are denoted with subscripts "$b$" and "$d$", respectively. Results marked with $\dagger$ are obtained from our experiments, while the remaining results are reported in (Xiao et al., 2024). Our proposed CaMIB achieves state-of-the-art performance, highlighted in bold.

| Methods | CMU-MOSI | | | | CMU-MOSEI | | | |
|---|---|---|---|---|---|---|---|---|
| | Acc2↑ | F1↑ | MAE↓ | Corr↑ | Acc2↑ | F1↑ | MAE↓ | Corr↑ |
| **BERT** | | | | | | | | |
| Self-MM$_b$ (Yu et al., 2021) | 84.0 | 84.4 | 0.713 | 0.798 | 85.0 | 85.0 | 0.529 | 0.767 |
| MMIM$_b$ (Han et al., 2021b) | 84.1 | 84.0 | 0.700 | 0.800 | 86.0 | 86.0 | 0.526 | 0.772 |
| MAG$_b$ (Rahman et al., 2020) | 86.1 | 86.0 | 0.690 | 0.831 | 84.8 | 84.7 | 0.543 | 0.755 |
| C-MIB$^\dagger_b$ (Mai et al., 2023c) | 85.2 | 85.2 | 0.728 | 0.793 | 86.2 | 86.2 | 0.584 | 0.789 |
| **DeBERTa** | | | | | | | | |
| Self-MM$_d$ (Yu et al., 2021) | 55.1 | 53.5 | 1.44 | 0.158 | 65.3 | 65.4 | 0.813 | 0.208 |
| MMIM$_d$ (Han et al., 2021b) | 85.8 | 85.9 | 0.649 | 0.829 | 85.2 | 85.4 | 0.568 | 0.799 |
| MAG$_d$ (Rahman et al., 2020) | 84.2 | 84.1 | 0.712 | 0.796 | 85.8 | 85.9 | 0.636 | **0.800** |
| C-MIB$^\dagger_d$ (Mai et al., 2023c) | 87.2 | 87.2 | 0.650 | 0.840 | 86.6 | 86.6 | 0.526 | 0.779 |
| ITHP$^\dagger_d$ (Xiao et al., 2024) | 88.2 | 88.2 | 0.654 | 0.844 | 86.2 | 86.3 | 0.556 | 0.781 |
| CaMIB | **89.8** | **89.8** | **0.616** | **0.857** | **87.3** | **87.3** | **0.517** | 0.788 |

compared to the BERT-based version. Despite this improvement, existing models—including MMIM and C-MIB—still lag behind our CaMIB model, indicating that simply adopting a stronger text encoder is not sufficient to reach state-of-the-art performance. CaMIB consistently achieves the highest results across all reported metrics. On CMU-MOSI, it reaches 89.8% Acc2 and F1, 0.616 MAE, and 0.857 correlation, outperforming all DeBERTa-based baselines. On CMU-MOSEI, it achieves 87.3% Acc2 and F1, 0.517 MAE, and 0.788 correlation, demonstrating robust generalization across datasets. These results highlight that CaMIB's design—integrating the information bottleneck for unimodal noise filtering, a parameterized mask generator for disentangling causal and shortcut components, and attention-based instrumental variable mechanisms—effectively captures causal multimodal features, rather than relying solely on stronger PLMs.

Moreover, the performance gaps between BERT- and DeBERTa-based models emphasize that while advanced PLMs provide a better textual foundation, the key contribution of CaMIB lies in its causal representation learning and debiasing mechanisms. This suggests that careful modeling of confounding factors and disentanglement of causal versus spurious signals is crucial for achieving state-of-the-art performance in MSA tasks. Overall, these findings demonstrate that CaMIB leverages both a powerful PLM backbone and sophisticated causal modeling techniques, resulting in superior performance, robustness, and generalization compared to existing baselines.

### B.3 ANALYSIS OF UNIMODAL AND BIMODAL SYSTEMS

In this subsection, we evaluate the performance of the unimodal, bimodal, and full multimodal variants of CaMIB on the CMU-MOSI and CMU-MOSEI datasets. Consistent with prior findings (Yang et al., 2023; Mai et al., 2023c; 2025b), the text modality remains the most informative source for sentiment prediction, while audio and visual modalities provide complementary cues. Therefore, our analysis emphasizes configurations where the text modality is used alone, in combination with one auxiliary modality, or in the full trimodal setting.

Table 4 summarizes the results across different modality combinations. Several key observations emerge: **i)** In all unimodal and bimodal configurations, CaMIB consistently outperforms the baseline ITHP on both datasets. Remarkably, in the text-only setting, CaMIB achieves an Acc7 of **50.4%** on CMU-MOSI, representing a substantial improvement over ITHP's 42.3%, and reaches 54.7% on CMU-MOSEI. Notably, this establishes a new state-of-the-art for seven-class classification with the text modality alone. Similarly, bimodal combinations such as text-audio and text-visual show marked improvements across all metrics, demonstrating the effectiveness of causal debiasing and information bottleneck mechanisms even when only partial modalities are available. **ii)** Across all

Table 4: Performance comparison of CaMIB and ITHP on CMU-MOSI and CMU-MOSEI across different modality combinations

| Methods | CMU-MOSI | | | | | CMU-MOSEI | | | | |
|---|---|---|---|---|---|---|---|---|---|---|
| | Acc7↑ | Acc2↑ | F1↑ | MAE↓ | Corr↑ | Acc7↑ | Acc2↑ | F1↑ | MAE↓ | Corr↑ |
| ITHP (Text) | 42.3 | 85.3/87.0 | 85.2/87.0 | 0.726 | 0.817 | 52.2 | 76.6/84.1 | 77.7/84.3 | 0.553 | 0.777 |
| CaMIB (Text) | **50.4** | **86.4/87.8** | **86.4/87.8** | **0.628** | **0.848** | **54.7** | **82.3/86.4** | **82.8/86.4** | **0.516** | **0.784** |
| ITHP (Text-Audio) | 46.7 | 84.8/86.7 | 84.8/86.7 | 0.656 | 0.841 | 53.3 | **85.4**/86.4 | **85.5**/86.2 | 0.522 | **0.786** |
| CaMIB (Text-Audio) | **47.5** | **86.7/88.2** | **86.7/88.2** | **0.644** | **0.844** | **54.4** | 82.4/**86.6** | 82.8/**86.5** | **0.517** | 0.782 |
| ITHP (Text-Visual) | 43.5 | 85.4/87.5 | 85.3/87.4 | 0.695 | 0.832 | 53.4 | **84.1/87.2** | **84.4/87.2** | 0.532 | **0.791** |
| CaMIB (Text-Visual) | **48.3** | **87.5/89.0** | **87.4/89.0** | **0.635** | **0.849** | **54.2** | 82.3/86.6 | 82.3/86.6 | **0.517** | 0.783 |
| ITHP (Full) | 46.3 | 86.1/88.2 | 86.0/88.2 | 0.654 | 0.844 | 51.6 | 82.3/86.2 | 82.9/86.3 | 0.556 | 0.781 |
| CaMIB (Full) | **48.0** | **88.2/89.8** | **88.1/89.8** | **0.616** | **0.857** | **53.5** | **85.3/87.3** | **85.4/87.2** | **0.517** | **0.788** |

metrics, bimodal configurations generally achieve higher performance than their unimodal counterparts, while the full trimodal models attain the best results. For example, the full CaMIB model achieves 89.8% Acc2 and 0.616 MAE on CMU-MOSI, and 87.3% Acc2 and 0.517 MAE on CMU-MOSEI, surpassing the corresponding ITHP variants. This trend underscores the value of integrating complementary modalities and reinforces the well-established benefits of multimodal fusion for sentiment analysis. **iii)** CaMIB demonstrates robustness in scenarios with missing modalities. Even when only one auxiliary modality is available alongside text, the performance remains consistently strong, with limited degradation compared to the full trimodal setting. This property highlights CaMIB's practical applicability in real-world conditions where inputs may be partially missing or noisy.

Overall, these results confirm that CaMIB effectively leverages causal representations and multimodal fusion to achieve robust, high-performance sentiment analysis, even with incomplete or corrupted modalities, thereby demonstrating both its flexibility and generalizability.

## B.4 ANALYSIS OF MODEL COMPLEXITY

Table 5: Comparison of the number of parameters between CaMIB with its variants and its baseline ITHP.

| Model | Number of Parameters |
|---|---|
| ITHP (Xiao et al., 2024) | 184, 883, 706 |
| w/o CaMIB | 185, 029, 441 |
| w/o IB | 188, 604, 040 |
| CaMIB | 189, 246, 280 |

Our CaMIB model is built upon the ITHP baseline, with several architectural enhancements specifically designed for causal representation learning. These include: i) an information bottleneck module that filters out unimodal noise irrelevant to prediction; ii) a parameterized mask generator that disentangles causal and shortcut components of the fused representation space; and iii) an instrumental variable mechanism with attention-based regularization to ensure global causal consistency.

As shown in Table 5, the baseline ITHP contains 184.9M parameters. Removing all CaMIB-related components (w/o CaMIB) reduces the model to the basic ITHP architecture, with nearly the same parameter count of 185.0M. In contrast, removing only the information bottleneck (w/o IB) while keeping the other CaMIB components increases the parameter size to 188.6M, reflecting the cost of disentanglement and attention mechanisms. The full CaMIB model reaches 189.2M parameters, corresponding to only a 2.3% increase over ITHP—a modest overhead given the substantial performance improvements.

This analysis indicates that most of the additional complexity arises from the disentanglement and attention-based causal modules, while the IB itself contributes little to parameter growth. Overall, CaMIB strikes a favorable balance, introducing causal modeling capabilities and robustness under distribution shifts while keeping the parameter overhead minimal.

## C  DATASETS INFORMATION

We evaluate the proposed CaMIB model on five benchmark datasets spanning three tasks: Multimodal Sentiment Analysis (MSA), Multimodal Humor Detection (MHD), and Multimodal Sarcasm Detection (MSD).

- **CMU-MOSI** (Zadeh et al., 2016): A widely used benchmark for MSA, comprising over 2,000 video utterances collected from online platforms. Each utterance is annotated with a sentiment intensity score on a seven-point Likert scale ranging from $-3$ (most negative) to 3 (most positive).

- **CMU-MOSEI** (Zadeh et al., 2018): One of the largest and most diverse datasets for MSA, containing more than 22,000 video utterances from over 1,000 YouTube speakers across approximately 250 topics. Each utterance is annotated with both categorical emotions (six classes) and sentiment scores on the same $-3$ to 3 scale as CMU-MOSI. In our experiments, we focus on sentiment scores to ensure consistency.

- **CMU-MOSI (OOD)** (Sun et al., 2022): An out-of-distribution (OOD) variant of CMU-MOSI, constructed via an adapted simulated annealing algorithm (Aarts et al., 1987) that iteratively modifies the test distribution. This process introduces substantial shifts in word–sentiment correlations compared to the training set, thereby providing a challenging benchmark for assessing model robustness under distribution shifts in MSA.

- **UR-FUNNY** (Hasan et al., 2019): A benchmark dataset for the MHD task, derived from TED talk videos featuring 1,741 speakers. Each target utterance, referred to as a punchline, is annotated across language, acoustic, and visual modalities. The utterances preceding the punchline serve as contextual inputs for the model. Punchlines are identified using the *laughter* tag in transcripts, which marks audience laughter; negative samples are similarly obtained when no laughter follows. The dataset is split into 7,614 training, 980 validation, and 994 testing instances. Following prior works (Hasan et al., 2021; Chen et al., 2025; Mai et al., 2025a), we adopt version 2 of UR-FUNNY in our experiments.

- **MUStARD** (Castro et al., 2019): A dataset designed for the MSD task, collected from popular television series such as *Friends*, *The Big Bang Theory*, *The Golden Girls*, and *Sarcasmaholics*. It comprises 690 video utterances manually annotated as sarcastic or non-sarcastic. Each instance includes both the target punchline utterance and its preceding dialogue to provide contextual information.

## D  IMPLEMENTATION DETAILS

### D.1  FEATURE EXTRACTION

**Text Modality:** For the MSA task, textual embeddings are obtained using DeBERTa (He et al., 2020), following the recent state-of-the-art approach (Xiao et al., 2024). For the MHD and MSD tasks, contextual word representations are derived from a pretrained BERT (Devlin et al., 2019) model.

**Acoustic Modality:** Acoustic features are extracted using COVAREP (Degottex et al., 2014), including 12 Mel-frequency cepstral coefficients, pitch, speech polarity, glottal closure instants, and the spectral envelope. Features are computed over the entire audio clip of each utterance, forming a temporal sequence that captures dynamic variations in vocal tone.

**Visual Modality:** For the MSA task, visual features are extracted with Facet (iMotions 2017, https://imotions.com/), including facial action units, landmarks, head pose, and other relevant cues. These features form a temporal sequence representing facial expressions over time. For the MHD and MSD tasks, following prior works (Hasan et al., 2021; Mai et al., 2025a), OpenFace 2 (Baltrušaitis et al., 2016) is used to extract facial action units as well as rigid and non-rigid facial shape parameters.

Table 6: Hyper-parameters of CaMIB. Notably, since the CMU-MOSI (OOD) dataset is divided into a seven-class bias dataset and a two-class bias dataset, it has two sets of hyperparameters. In our work, we only needed to adjust the disentanglement parameter to achieve state-of-the-art performance.

| Hyper-parameter | CMU-MOSI | CMU-MOSEI | UR-FUNNY | MUStARD | CMU-MOSI (OOD) |
|---|---|---|---|---|---|
| Batch Size | 8 | 32 | 256 | 64 | 8 |
| Epochs | 30 | 15 | 20 | 10 | 30 |
| Warm-up | ✓ | ✓ | ✓ | ✓ | ✓ |
| Initial Learning Rate | $1 \times 10^{-5}$ | $1 \times 10^{-5}$ | $2 \times 10^{-5}$ | $7 \times 10^{-5}$ | $1 \times 10^{-5}$ |
| Optimizer | AdamW | AdamW | AdamW | AdamW | AdamW |
| Dropout Rate | 0.5 | 0.1 | 0.6 | 0.5 | 0.5 |
| Fusion Feature Dimension $d$ | 512 | 512 | 128 | 256 | 512 |
| Disentanglement Loss Weight $\lambda_1$ | 0.2 | 1.0 | 0.1 | 0.8 | 0.2 / 1.0 |
| Causal Intervention Loss Weight $\lambda_2$ | 0.3 | 0.7 | 0.2 | 0.3 | 0.9 |
| Information Bottleneck Loss Weight $\beta$ | 1e-4 | 1e-2 | 1e-5 | 1e-4 | 1e-4 |

## D.2 EXPERIMENTAL DETAILS

We implement the proposed CaMIB model using the PyTorch framework on an NVIDIA RTX A6000 GPU (48GB) with CUDA 11.6 and PyTorch 1.13.1. The training process employs the AdamW optimizer (Loshchilov & Hutter, 2017). Detailed hyperparameter settings are provided in Table 6. To identify the optimal configuration, we conduct a comprehensive grid search with fifty random iterations. The batch size is selected from $\{8, 16, 32, 64, 128, 256\}$, while the initial learning rate and fusion feature dimension are searched over $\{1e-5, 2e-5, 4e-5, 7e-5, 9e-5\}$ and $\{64, 128, 256, 512\}$, respectively. The dropout rate is chosen from $\{0.1, 0.2, 0.3, 0.4, 0.5, 0.6\}$, and the hyperparameters $\lambda_1$ and $\lambda_2$ are tuned within $\{0.1, 0.2, \ldots, 1.0\}$, while $\beta$ is searched over $\{1, 1e-1, 1e-2, 1e-3, 1e-4, 1e-5\}$. Other hyperparameters are kept at predefined values. The final selection is based on the set that achieves the lowest MAE on the validation set.

## E EVALUATION METRICS

We evaluate the model's performance on the MSA task using a set of well-established metrics, reported for both CMU-MOSI and CMU-MOSEI datasets. For interpretability, classification results are presented as percentages. These metrics are calculated as follows:

**Seven-category Classification Accuracy (Acc7):** Measures the model's ability to predict fine-grained sentiment categories by dividing the sentiment score range ($-3$ to $3$) into seven equal intervals. The metric is defined as:

$$\text{Acc7} = \frac{1}{n} \sum_{i=1}^{n} \mathbf{1}(\hat{c}_i = c_i), \tag{49}$$

where $c_i$ and $\hat{c}_i$ denote the ground-truth and predicted categories of sample $i$, respectively, and $\mathbf{1}(\cdot)$ is the indicator function. Higher values indicate better fine-grained sentiment classification.

**Binary Classification Accuracy (Acc2):** Reflects the proportion of correct predictions in binary sentiment classification. For the MSA task, following prior works (Han et al., 2021b; Yang et al., 2023; Zhang et al., 2023; Wu et al., 2025b), we report two configurations: (i) *Negative/Non-negative (including zero)*: distinguishes negative sentiments ($< 0$) from non-negative sentiments ($\geq 0$); (ii) *Negative/Positive (excluding zero)*: focuses on strictly negative ($< 0$) versus positive ($> 0$) sentiments. The metric is formulated as:

$$\text{Acc2} = \frac{TP + TN}{TP + TN + FP + FN}, \tag{50}$$

where $TP$, $TN$, $FP$, and $FN$ denote true positives, true negatives, false positives, and false negatives, respectively.

**Weighted F1-score (F1):** Computes the harmonic mean of precision and recall while considering class-specific weights to mitigate imbalance. It is formulated as:

$$\text{F1} = 2 \cdot \frac{Precision \cdot Recall}{Precision + Recall}, \tag{51}$$

where $\text{Precision} = \frac{TP}{TP+FP}$ and $\text{Recall} = \frac{TP}{TP+FN}$.

**Mean Absolute Error (MAE):** Represents the average magnitude of prediction errors with respect to the ground-truth sentiment scores. It directly corresponds to the original sentiment scale, making it both intuitive and informative:

$$\text{MAE}(\hat{y}, y) = \frac{1}{n} \sum_{i=1}^{n} |\hat{y}_i - y_i|, \tag{52}$$

where $y_i$ is the true label, $\hat{y}_i$ is the predicted value, and $n$ is the total number of predictions.

**Pearson Correlation Coefficient (Corr):** Quantifies the strength and direction of the linear relationship between predicted and true sentiment scores:

$$\text{Corr}(x, y) = \frac{\sum_{i=1}^{n}(x_i - \bar{x})(y_i - \bar{y})}{\sqrt{\sum_{i=1}^{n}(x_i - \bar{x})^2}\sqrt{\sum_{i=1}^{n}(y_i - \bar{y})^2}}, \tag{53}$$

where $x_i$ and $y_i$ denote predicted and ground-truth values, respectively, and $\bar{x}, \bar{y}$ are their means.

For CMU-MOSI under the OOD setting, we follow prior causality-based approaches (Sun et al., 2022; 2023; Huan et al., 2024; Jiang et al., 2025) and report only classification metrics for fair comparison. Similarly, for the MHD and MSD tasks, in line with prior methodologies (Hasan et al., 2021; Huang et al., 2025; Mai et al., 2025a), we report only binary accuracy, which evaluates the model's ability to distinguish between humorous and non-humorous, as well as sarcastic and non-sarcastic, instances.

# F    BASELINES

We compare CaMIB against the following twenty-five representative baselines. Note that, due to differences in task settings, we select different subsets of these baselines for each specific task.

1. **MulT** (Tsai et al., 2019): Multimodal Transformer (MulT) constructs multimodal representations by leveraging cross-modal Transformers to map information from source modalities into target modalities.

2. **MISA** (Hazarika et al., 2020): Modality-Invariant and -Specific Representation (MISA) maps unimodal features into two separate embedding subspaces for each modality, distinguishing between modality-specific and modality-invariant information.

3. **MAG** (Rahman et al., 2020): Multimodal Adaptation Gate (MAG) employs an adaptation gate that allows large pre-trained transformers to integrate multimodal information during fine-tuning.

4. **Self-MM** (Yu et al., 2021): Self-Supervised Multi-task Multimodal (Self-MM) sentiment analysis network uses annotated global sentiment labels to create pseudo labels for individual modalities, thereby guiding the model to acquire discriminative unimodal representations.

5. **MMIM** (Han et al., 2021b): MultiModal InfoMax (MMIM) maximizes mutual information both among unimodal representations and between multimodal and unimodal representations, promoting the learning of richer multimodal features.

6. **BBFN** (Han et al., 2021a): Bi-Bimodal Fusion Network (BBFN) performs simultaneous fusion and separation on pairwise modality representations, using a gated Transformer to handle modality imbalance.

7. **HKT** (Hasan et al., 2021): Humor Knowledge enriched Transformer (HKT) integrates context and humor-centric external knowledge to capture multimodal humorous expressions using Transformer-based encoders and cross-attention.

8. **HyCon** (Mai et al., 2022): Hybrid Contrastive Learning (HyCon) integrates intra-modal and inter-modal contrastive learning to model interactions both within individual samples and across different samples or categories.

9. **MIB** (Mai et al., 2023c): Multimodal Information Bottleneck (MIB) utilizes the information bottleneck principle to suppress redundancy and noise in both unimodal and multimodal representations.

10. **AOBERT** (Kim & Park, 2023): All-modalities-in-One BERT (AOBERT) is a single-stream Transformer pre-trained with multimodal masked language modeling and alignment prediction to capture intra- and inter-modality relationships.

11. **MCL** (Mai et al., 2023a): Multimodal Correlation Learning (MCL) is an architecture designed to capture correlations across modalities, enhancing multimodal representations while preserving modality-specific information.

12. **MGCL** (Mai et al., 2023b): Multimodal Global Contrastive Learning (MGCL) learns multimodal representations from a global view using contrastive learning with permutation-invariant fusion and label-guided positive/negative sampling.

13. **ConFEDE** (Yang et al., 2023): Contrastive FEature DEcomposition (ConFEDE) conducts contrastive representation learning in conjunction with contrastive feature decomposition to enhance multimodal representations.

14. **DMD** (Li et al., 2023): Decoupled Multimodal Distillation (DMD) enhances emotion recognition by decoupling each modality into modality-relevant and modality-exclusive spaces and performing adaptive cross-modal knowledge distillation via a dynamic graph.

15. **KuDA** (Feng et al., 2024): Knowledge-Guided Dynamic Modality Attention (KuDA) adaptively selects the dominant modality and adjusts modality contributions using sentiment knowledge for multimodal sentiment analysis.

16. **ITHP** (Xiao et al., 2024): Information-Theoretic Hierarchical Perception (ITHP), grounded in the information bottleneck principle, designates a primary modality while using other modalities as detectors to extract salient information.

17. **DLF** (Wang et al., 2025): Disentangled-Language-Focused (DLF) separates modality-shared and modality-specific features, employs geometric measures to minimize redundancy, and utilizes a language-focused attractor with cross-attention to strengthen textual representations.

18. **DEVA** (Wu et al., 2025b): DEVA generates textual sentiment descriptions from audio-visual inputs to enhance emotional cues, and employs a text-guided progressive fusion module to improve alignment and fusion in nuanced emotional scenarios.

19. **MOAC** (Mai et al., 2025a): Multimodal Ordinal Affective Computing (MOAC) enhances affective understanding by performing coarse-grained label-level and fine-grained feature-level ordinal learning on multimodal data.

**In particular, we include six causality-based baselines:**

20. **CLUE** (Sun et al., 2022): CounterfactuaL mUltimodal sEntiment (CLUE) employs causal inference and counterfactual reasoning to remove spurious direct textual effects, retaining only reliable indirect multimodal effects to enhance out-of-distribution generalization.

21. GEAR (Sun et al., 2023): General dEbiAsing fRamework (GEAR) separates robust and biased features, estimates sample bias, and applies inverse probability weighting to down-weight highly biased samples, thereby improving out-of-distribution robustness.

22. **MulDeF** (Huan et al., 2024): Multimodal Debiasing Framework (MulDeF) employs causal intervention with frontdoor adjustment and multimodal causal attention during training, and leverages counterfactual reasoning at inference to eliminate verbal and nonverbal biases, thereby enhancing out-of-distribution generalization.

23. **AtCAF** (Huang et al., 2025): Attention-based Causality-Aware Fusion (AtCAF) captures causality-aware multimodal representations using a text debiasing module and counterfactual cross-modal attention for sentiment analysis.

24. **AGS-SMoE** (Chen et al., 2025): Adaptive Gradient Scaling with Sparse Mixture-of-Experts (AGS-SMoE) mitigates modal preemption by dynamically scaling gradients and using sparse experts to balance multimodal optimization.

25. **MMCI** (Jiang et al., 2025): Multi-relational Multimodal Causal Intervention (MMCI) models multimodal inputs as a multi-relational graph and applies backdoor adjustment to disentangle causal and shortcut features for robust sentiment analysis.

