# OpenReview forum: "Towards Minimal Causal Representations for Human Multimodal Language Understanding"
_ICLR.cc/2026/Conference — Submitted to ICLR 2026_

### Official Review · Reviewer_C3c7 · 2025-10-31

**Soundness:** 3
**Presentation:** 3
**Contribution:** 3
**Rating:** 6
**Confidence:** 3

**Summary:**

The paper targets the problem that MLU models pick up spurious shortcuts and fail under distribution shifts. It proposes CaMIB: first apply an Information Bottleneck to each modality to filter noise, then fuse features and use a learnable mask to split them into causal and shortcut subrepresentations, guided by an instrumental variable built with self-attention to capture global inter-modal and token-level dependencies; finally, use a backdoor adjustment by randomly recombining causal and shortcut features so the model relies on causal information. Experiments on sentiment, humor, and sarcasm (including OOD tests) show better robustness and accuracy than prior methods. The work contributes an SCM for MLU that treats redundancy-induced spurious correlations as confounders, and a simple end-to-end framework that improves interpretability and OOD generalization.

**Strengths:**

The paper proposes a task-agnostic Structural Causal Model for MLU that treats redundancy-induced spurious correlations as confounders, rather than limiting attention to predefined bias types. Building on this causal view, it introduces an attention-derived instrumental variable to anchor causal signals, a learnable mask that splits fused multimodal features into causal vs. shortcut parts, and a backdoor “remix” intervention at the representation level to break residual shortcut reliance. This combination is a fresh, creative, and lightweight approach to multimodal debiasing, and is broadly applicable across MLU tasks.

Robustness to distribution shift is central in multimodal learning. By operationalizing causal principles with simple modules compatible with modern encoders, the approach is practical.

**Weaknesses:**

1. The “with IV” model includes strong cross-modal global attention; if “w/o IV” lacks a capacity-matched global dependency module, the ablation conflates “having IV” with “having any global dependency modeling,” making the comparison unfair.
2. The alignment loss ||Zc−V||^2 encourages Zc to align with V; however, if V contains environment or shortcut information, the mask might be "misled," causing the model to treat spurious correlations as causal relationships. This could undermine the validity of the causal separation.
3. The current “w/o INTV” ablation only sets the intervention weight to zero, which conflates removing a loss term with removing the intervention mechanism itself. It does not test a setting where the model relies solely on the causal path without any random recombination with shortcut features, so it’s unclear whether the robustness gain comes from the intervention operation or simply from using the causal representation.

**Questions:**

1. The current ablation conflates “having IV” with “having any global cross-modal/token-level dependency modeling.” To know what truly drives the gains, you need to isolate the effect of dependencies from the IV alignment itself under capacity-matched settings.
(1) Build V via cross-modal, token-level attention but do not use the IV alignment constraint.
(2) Keep the IV alignment constraint, but replace the attention-built V with a capacity-matched non-attention IV (e.g., SimpleSum / MeanPool, Max/Avg pooling).
2. How do you ensure that V does not contain environment or shortcut information that could bias the mask towards spurious correlations?
3. Please add a Pure-causal (no recombination) ablation: use only the causal representation for prediction and do not perform any random recombination with shortcut features; keep all other modules and settings the same.

---

### Official Review · Reviewer_n8dy · 2025-11-01

**Soundness:** 1
**Presentation:** 2
**Contribution:** 1
**Rating:** 2
**Confidence:** 4

**Summary:**

The draft tries to do a disentanglement of content and noise for the OOD problem and base on the fact that if we can only use content to predict the final label the model would be much more reliable. However, there are several technical mistakes in the draft. First of all, in the causal graph of Figure 1 (b), the authors claims that they cut the edges between Z and C by enforce the independency of them. Unfortunately this is not enough to recover Z and C causally. Basically Z and C are latent variables and the mapping between [Z, C] and M are nonlinear. In that case, by the theory of non-linear ICA, in general Z and C are not identifiable. On simple example is that if we assume that both Z and C are one dimensional, and let [Z, C] \sim N([0, 0], [[1, 0], [0, 1]], and we can see that Z and C are independent. However, we can apply any orthogonal transformation on [Z, C] to get a new [\hat Z, \hat C] and then they will be \hat Z and \hat C will also be independent, but however \hat C is a mixture of Z and C.

Since Z and C both contains information about Y, in general the label Y will also not help in this procedure if we do not have further assumptions. Based on those facts, the basic model in the draft is unfortunately wrong.

Furthermore, in the theoretical analysis, it seems that the authors assume that the attention score do have a causal meaning, and this is in general not correct.

**Strengths:**

As the basic model in the paper is wrong, as well as some other mistakes in the draft, the only strength I found is the experimental results.

**Weaknesses:**

1. The basic model in the draft is wrong: enforce the independence between C and Z is not sufficient for block disentanglement, further assumption or constraints are required;

2. Some mistakes in concepts:  e.g. line 154-155, ``Link M→Y: existing MLU methods predict directly from M, potentially introducing spurious correlations caused by Z.`` This is incorrect, estimate Y directly from M does not mean that spurious correlations will be introduced. In the inference stage, we always have to inference Y using M as the prediction of Y can be viewed as an inverse problem.

**Questions:**

See above summary and weakness.

---

### Official Review · Reviewer_dw7V · 2025-11-02

**Soundness:** 2
**Presentation:** 2
**Contribution:** 2
**Rating:** 2
**Confidence:** 5

**Summary:**

The paper proposes CaMIB, a causal, training-time framework for multimodal language understanding (MLU) that aims to disentangle causal vs. shortcut factors in fused representations to improve OOD generalization. The Information Bottleneck (IB) filtering compresses each unimodal stream (text/audio/visual) to remove task-irrelevant noise. The self-attention–derived instrumental variable (IV) captures global inter-/intra-modal dependencies.

**Strengths:**

* Clear SCM, IB filtering, IV via attention, mask-based disentanglement, and backdoor recombination form a coherent recipe that targets shortcut paths in multimodal fusion.

* Competitive or SOTA results on MSA, humor, sarcasm, and OOD MOSI, plus ablations and sensitivity analyses (λ₁, λ₂, β) that substantiate each module’s role.

**Weaknesses:**

* The self-attention–constructed IV may still correlate with shortcuts; the paper does not rigorously test IV exogeneity/strength or provide diagnostics beyond performance gains.

*  Learning a soft mask to split Z into (Zc, Zs) without supervision risks entanglement, posterior collapse, or leakage; guarantees of identifiability are limited to heuristic losses.

* Forcing a uniform predictive head may over-regularize informative but non-causal cues and degrade uncertainty calibration; no calibration metrics are reported.

* Experiments center on small/medium MLU benchmarks with pre-extracted features; compute/memory overhead of IB encoders, attention-based IV, mask generator, and intervention sampling is not quantified in detail.

**Questions:**

* How do you verify the instrumental variable is (approximately) exogenous and strong? Can you report IV–shortcut correlations, partial R², or leave-one-modality stress tests to support the causal story?

* What prevents Zc/Zs role-swaps or leakage? Have you tried sparsity/entropy constraints, orthogonality penalties, or causal supervision (e.g., known spurious cues) to stabilize disentanglement?

* How does CaMIB affect ECE/Brier on in-/OOD splits? Does pushing Zs toward uniform predictions improve or hurt calibration and selective prediction?

* What are the training/inference costs (GPU hours, peak memory) and the overhead attributable to IB, IV attention, mask generation, and intervention sampling? How does CaMIB scale to longer sequences or more modalities?

---

### Official Review · Reviewer_r765 · 2025-11-04

**Soundness:** 3
**Presentation:** 3
**Contribution:** 3
**Rating:** 6
**Confidence:** 3

**Summary:**

This paper proposes the **Causal Multimodal Information Bottleneck (CaMIB)** to address poor out-of-distribution (OOD) generalization in Human Multimodal Language Understanding (MLU) (caused by confusing statistical shortcuts with causal features). CaMIB combines the Information Bottleneck (IB) for unimodal noise filtering, a mask generator to disentangle causal/shortcut representations, instrumental variables for global causal consistency, and backdoor adjustment for stable estimation. It outperforms baselines on sentiment analysis, humor/sarcasm detection, and OOD tasks.

**Strengths:**

1. It innovatively integrates IB and causal inference to solve OOD generalization, filling the gap between statistical compression and causal robustness in MLU.

2. Its design (e.g., Structural Causal Model, instrumental variables) is theoretically rigorous, supported by detailed derivations to ensure causal soundness.

3. It undergoes comprehensive validation across 3 MLU tasks, 5 datasets, and OOD settings, with ablation studies confirming component effectiveness.

**Weaknesses:**

1. It shows high sensitivity to hyperparameters (λ₁, λ₂) under OOD scenarios, increasing deployment complexity.
2. It lacks detailed analysis of inference speed or computational cost, limiting assessment for latency-sensitive applications.

**Questions:**

Please refer to Weaknesses.

---

### Meta-Review · Area_Chair_5sew · 2026-01-06

**Summary:**

This paper proposes a causal framework for multimodal language understanding. Several reviewers find the problem important and the overall framework coherent, and acknowledge the competitive empirical results.

However, multiple reviewers (n8dy, dw7V, C3c7) raise fundamental concerns about the soundness of the causal formulation, particularly regarding the identifiability of latent variables, the rigor of the proposed method and its claimed causal properties, and the validity of the proposed instrumental variable. The fairness of some experimental comparisons is questioned (C3c7). While the empirical performance is recognized, these issues go to the core of the paper’s theoretical claims.

No rebuttal was provided to address these concerns. Based on the reviewers’ assessments, the AC recommends rejection.

**Reviewer Concerns:**

No rebuttal was provided to address reviewers’ concerns. The outstanding issues relate to fundamental questions about the soundness of the causal formulation, particularly the identifiability of latent variables, the rigor of the proposed method and its claimed causal properties, and the fairness of the experimental comparisons also tied to the theoretical assumptions.

**Reviewer Scores:**

The original scores are 6,6,2,2. No rebuttal was provided. The score of each reviewer would likely be lower than the original scores or maintained as original.

---

### Decision · Program_Chairs · 2026-01-26

Reject